# Defining or distinguishing Self-supervised Learning and Weakly Supervised Learning

## Abstract

The AI community has been a very rapidly growing community producing a vast amount of research in a very short span of time. These researches generate a lot of new methods and terminologies. With this scale of developments, it is very difficult to keep a track of terminologies, and under such conditions even more difficult to standardize the definitions. In the wake of such scenarios, we try to perform a detailed study of some terminologies in representation learning and form standard definitions of them. With this work, we establish a clear distinction between the concepts of Self-supervised learning and Weakly supervised learning.

## 1 Introduction

The AI research community is currently thriving with new methodologies and techniques being investigated everyday. Keeping track of all these advancements can be very challenging.

In this work, we try to compare two popular terminologies that are being used interchangeably by the research community. We identify the works that have tried to define one or both terminologies in a certain way and critique the works that define them differently. In the end, we try to define the terminologies so that they make a clear distinction between them.

## 2 Definitions

We compare two popular and fast-emerging methods, namely, Self-Supervised Learning and Weakly-Supervised Learning.

### 2.1 Self-Supervised Learning

Semi-supervised Learning (SSL) has been considered as a mid-way between supervised and unsupervised learning. Some research explains self-supervised learning as the method that uses unsupervised learning based pretext tasks that can be used for downstream tasks Huang et al. (2021); Purushwalkam & Gupta (2020); Zhai et al. (2019). Self-supervised learning methods do not involve the use of any manual labels Liu et al. (2023a).

The other works claim the method contains unlabeled data along with some supervision information-but not necessarily for all examples, hence claiming the method to learning from large amount of unlabled data and small amount of labeled data Chapelle et al. (2009); Ouali et al. (2020) Zhu (2005) Jaiswal et al. (2021) Jing & Tian (2019). Hence, some works classify self-supervised learning to be a part of weakly supervised learning Hernández-González et al. (2016). Semi-supervised learning attempts to build better classifiers using vast amounts of unlabeled data and small amounts of labeled data.

### 2.2 Weakly-Supervised Learning

Zhou (2017) refer to Weakly supervised learning as all the methods that attempt to conduct predictive models with weak supervision. Hernández-González et al. (2016) and Hernández-González & Pérez (2022) define Weakly supervised learning as learning from different kind of partially labeled data. Weak supervision is further classified into 3 subclasses Zhou (2017):

1. **Incomplete Supervision**: Subset of data is labeled while large set remains unlabeled.
2. **Inexact Supervision** : Only coarse grained labels are provided.
3. **Inaccurate Supervision**: Given labels are not always the ground truth.

Kumar & Rowley (2007) and Hernández-González & Pérez (2022) refer to weakly-labeled data as input data in terms of groups where data in each group belong to the same class, but individual class labels are not available for each data point.

## 3   The Ambiguity and Discussions

We clearly notice the ambiguity in the two definitions put in various research articles by different researchers. This has led to the usage of the terms Self-Supervised learning and Weakly-supervised learning interchangeably. We see that self-supervised learning has been defined as learning with a vast amount of unlabeled data and a small amount of labeled data which is also the same as "Incomplete Supervision" of Weakly supervised learning.

The definition of Incomplete supervision seems more apt with incompletely labeled datasets over self-supervised learning. Self-supervised learning, on the other hand, does not need any manual labels Liu et al. (2023b); Jaiswal et al. (2021); Misra & Maaten (2020); Hendrycks et al. (2019). Owing to the fact that self-supervised learning methods only use unlabeled data to generate representations and doesn't need any supervision of any kind, it would not be apt to categorize it under weakly supervised paradigm.

## 4   Our Definitions

After a detailed review of the existing literature, we put forward our own definitions which make a clear distinction between the two methods.

**Self-Supervised Learning**: We define self-supervised learning where the model learns by generating its own labels or tasks for supervision from unlabeled data.

**Weakly supervised learning**: We define weakly supervised learning as learning from weakly labeled data where the annotations are not complete or noisy. The model must rely on partial or indirect supervision to learn from the data.

In other words, self-supervised learning uses the data itself to generate supervision signals, while weakly supervised learning uses annotations that are not as precise or comprehensive as the ones in fully supervised learning.

## 5   Conclusion

Through this work, we attempt to give both the two popular definitions a more adequate consideration. We aim to promote a uniform interpretation of this concept and hence are trying to establish a precise definition. This would lead to an improved reliability and validity of research results. We hope to reduce misinterpretations and anticipate greater progress in research within the respective fields.

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
