# OpenReview forum: "Discerning  Self-Supervised Learning and Weakly Supervised Learning"
_ICLR.cc/2023/TinyPapers — Submitted to Tiny Papers @ ICLR 2023_

### Official Review · Reviewer_Umjg · 2023-03-19

**Confidence:** 5

**Summary Of Contributions:**

This paper reviewed and discussed the previous definitions on self-supervised learning and weakly-supervised learning. The authors the propose their definitions on them.

**Rating:**

Needs Clarification (NC): a submission which does not meet the reviewing criteria and needs clarification for its described problem or solution

**Strengths And Weaknesses:**

Strength:
The paper provides a comparison of two popular training strategies, namely self-supervised learning (SSL) and weakly supervised learning (WSL), and discusses the distinction between them, which can be useful for researchers and practitioners.

Weakness:
1. The main weakness is the motivation of the paper: I think the concepts/definitions of SSL and WSL learning are well-established and there is little confusion about them.
- For example, in sec 2.1, the authors discuss two interpretations of SSL. However, they do not conflict. SSL usually follows a two-stage pretrain-finetune process. In the first stage, the pretext tasks are designed to generate labels from the data itself to pretrain the model. In the second stage, a small amount of labeled data are used to finetune the model or simply train a linear readout for the downstream tasks, e.g. classification. The value of SSL lies in the good initialization of the model weight, so that in the second stage, the model can achieve comparable performance with the fully-supervised method using a small number of labels.
- Given the above, there actually won't be confusion between SSL and WSL, because the incomplete/noisy/inexact labels of WSL are from manual annotations (e.g. using object class labels for the semantic segmentation task) instead of generated from the data itself.

Minor:
The reference style is somehow distracting and I can't clearly parse which reference corresponds to which sentence. Suggestions: use a bracket (\citep{}) for all references except those at the beginning of the sentence. Also, it would be helpful to use hyperlinks to highlight (different colors) and references.

**Suggested Changes:**

General suggestions for better presentation:
- The addition of more visual aids or examples would enhance the paper and make the concepts more accessible to readers.
- People are usually very careful when trying to make definitions of some terminology. Mathematical equations are needed to rigorously define a concept. And it is dangerous to redefine the concepts without a thorough understanding and consideration of the whole field.

For more interesting future work:
- I suggest the authors really do some experiments using SSL and WSL on some datasets to better understand the definitions. Naturally, the strengths and weaknesses of SSL and WSL beyond the definitions can be then analyzed based on the experimental results.
- The authors may also consider exploring real-world scenarios and applications where SSL or WSL may be more advantageous to each other.

---

### Official Review · Reviewer_38Nw · 2023-03-31

**Confidence:** 4

**Summary Of Contributions:**

An attempt is made to understand and distinguish the concepts and literature around two closely related terminologies, namely Self-supervised learning and Weakly supervised learning.

**Rating:**

Great Start (GS): a submission which meets some of the reviewing criteria but has room for improvement

**Strengths And Weaknesses:**

Strengths
- A good study of terminology as used in the recent literature is provided.
- There is a chance of confusion, and risk of miscommunication in the literature in the absence of well-defined terminology.

Weaknesses
- The paper reads more as an educated commentary/summary than a precise research contribution.
- It is not clear if the goal of precise distinction between the two terminologies is achieved.

**Suggested Changes:**

There is still a lot of potential for ambiguity in the presented definitions. For example, what does the process of "generating its own labels or tasks" look like? Does it involve some kind of inductive bias? Where do "coarse grained labels" fit in?

I recommend trying to write the definitions mathematically. For example, Lee et al. NeurIPS 2021 for "self-supervised learning" and some references in the survey Zhou (2017) that you cite for "weakly supervised" learning could be useful starting points.

Also, in my opinion, it is not very uncommon to have terms having overlapping definitions (e.g. artificial intelligence vs. machine learning), which are not identical, and often very hard to settle debates on how such terms should be distinguished.

Typo:
- Section 2.1: Semi-supervised Learning --> Self-supervised Learning. "Semi-supervised Learning" actually corresponds to "Incomplete Supervision"

---

### Meta-Review · Area_Chair_jtAn · 2023-04-02

**Recommendation:** Invite to revise
**Confidence:** 5

**Metareview:**

Pros:
- distinction between popular and closely related terminologies can be useful for researchers and practitioners.
- usage of the terms in some recent literature is tracked

Cons:
- There is some challenge to the premise of the paper, the two terminologies considered seem to be clear from the literature
- The discussion itself is not mathematically precise and leaves open room for ambiguity.

**Summary:**

The authors attempt to distinguish the concepts of Self-supervised learning and Weakly supervised learning, but have missed mathematical definitions and do not provide any illustrative examples.

**Comments And Feedback To The Authors:**

Examples and experiments to distinguish SSL and WSL are recommended. Also it would be good to compare mathematical formulations of the problems to be more precise.

**Reason For Not Giving A Higher Recommendation:**

Reviewers concur with confidence.

**Reason For Not Giving A Lower Recommendation:**

N/A

---

### Decision · Program_Chairs · 2023-04-08

No revision received; not invited to archive